



**On the barium - oxygen consumption relationship in the Mediterranean Sea: implications**
**for mesopelagic marine snow remineralisation.**
Stéphanie H.M. Jacquet[1*], Dominique Lefèvre[1], Christian Tamburini[1], Marc Garel[1], Frédéric
A.C. Le Moigne[1], Nagib Bhairy[1], Marie Roumagnac[1], Sophie Guasco[1]
[1]Aix Marseille Université, CNRS/INSU, Université de Toulon, IRD, Mediterranean Institute of
Oceanography (MIO), UM 110, 13288 Marseille, France
[*]Correspondence to: S. Jacquet (*stephanie.jacquet@mio.osupytheas.fr*)



**ABSTRACT**
In the ocean, remineralisation rate associated with sinking particles is a crucial variable. Since
the 90's, particulate biogenic barium ($Ba_{xs}$) has been used as an indicator of carbon
remineralization by applying a transfer function relating $Ba_{xs}$ to $O_2$ consumption (Dehairs's
transfer function, Southern Ocean-based). Here, we tested its validity in the Mediterranean Sea
(ANTARES / EMSO-LO) for the first time by investigating connections between $Ba_{xs}$,
prokaryotic heterotrophic production (PHP) and oxygen consumption ($JO_2$-Opt; optodes
measurement). We show that: (1) higher $Ba_{xs}$ (409 pM; 100- 500 m) in situations where
integrated PHP (PHP100/500= 0.90) is located deeper, (2) higher $Ba_{xs}$ with increasing $JO_2$-Opt,
and (3) similar magnitude between $JO_2$-Opt (3.14 mmol $m^{-2}$ $d^{-1}$; 175- 450 m) and $JO_2$-Ba (4.59
mmol $m^{-2}$ $d^{-1}$; transfer function). Overall, $Ba_{xs}$, PHP and $JO_2$ relationships follow trends observed
in the Southern Ocean. We believe that such transfer function could apply in the Mediterranean
Sea with no restriction.


**KEY WORDS:** particulate biogenic barium, mesopelagic zone, oxygen consumption,
prokaryotic heterotrophic production, carbon remineralization, Mediterranean Sea





## 1. INTRODUCTION


Ocean ecosystems play a critical role in the Earth's carbon (C) cycle [IPCC, 2014]. The
quantification of their impacts of both present conditions and future predictions remains one of
the greatest challenges in oceanography [Siegel et al., 2016]. In essence, the biological C pump is
termed for the numerous processes involved in maintaining the vertical gradient in dissolved
inorganic C. This includes processes such as organic matter production in surface, its export and
subsequent remineralization. Most of marine snow organic C conversion (i.e. remineralization)
into $CO_2$ by heterotrophic organisms (i.e. respiration) occurs in the mesopelagic zone (100-1000
m) [Martin et al., 1987; Buesseler and Boyd, 2009]. Globally, the flux of C exported below 1000
m depth is the key determinant of ocean carbon storage capacity [Henson et al., 2011]. However,
there is no consensus on C transfer efficiency estimations from field experiments, leading to an
imbalance of the water column C budget [Giering et al., 2014]. Resolving this imbalance is in the
core of numerous studies in the global ocean, but also regionally, especially in the Mediterranean
Sea (MedSea). Due to limited exchanges with adjacent basin and the existence of an intense
overturning circulation qualitatively resembling the global one (but with shorter time scales), the
MedSea is often considered as a laboratory to observe and understand the impact of transient
climate variability on ecosystems and biogeochemical cycles [Malanotte-Rissoli et al., 2014]. In a
context of climate changes, better constraining C fluxes and the ocean C storage capacity is of
crucial importance.
Particulate barium in excess ($Ba_{xs}$, i.e. biogenic Ba from total particulate Ba after correction for
lithogenic Ba) is a geochemical tracer of particulate organic carbon (POC) remineralization in the
mesopelagic layer [Dehairs et al., 1997]. $Ba_{xs}$ mostly occurs in the form of barite microcrystals
($BaSO_4$) at these depths. In a global ocean undersaturated with respect to barite, studies report
that $Ba_{xs}$ would precipitate inside oversaturated biogenic micro-environments during POC



degradation by heterotrophic prokaryotes in the mesopelagic zone, through sulfate and/or barium
enrichment [Bertram and Cowen, 1997]. The first-ever studies on mesopelagic $Ba_{xs}$ reported
coinciding $Ba_{xs}$ maxima with depths of dissolved $O_2$ minimum and $pCO_2$ maximum [Dehairs et
al., 1987, 1997]. By using an 1D advection-diffusion model applied to highly resolved, precise
$O_2$ profiles in the Atlantic sector of the Southern Ocean (ANTX/6 cruise; Shopova et al., 1995),
Dehairs et al. [1997] established an algorithm converting mesopelagic $Ba_{xs}$ concentration into $O_2$
consumption rate ($JO_2$) and organic C remineralized (POC remineralization rate). This transfer
function has been widely used until now [Cardinal et al., 2001- Lemaitre et al., 2018]. Yet its
validity has never been tested in other oceanic provinces. Recently, significant progresses were
made in relating $Ba_{xs}$, $O_2$ dynamics to prokaryotic heterotrophic activity [Jacquet et al., 2015].
Nevertheless, the Dehairs transfer function has never been revised since. These advancements
clearly show that $Ba_{xs}$ is closely related with the vertical distribution of prokaryotes heterotrophic
production (PHP) (the rate of change with depth), reflecting the temporal progression of POC
remineralization processes. Also, in a first attempt to test the validity of the Dehairs's transfer
function in other locations, Jacquet et al. [2015] confronted oxygen consumption rates ($JO_2$) from
direct measurements (dark community respiration, DCR) to derived $JO_2$ from $Ba_{xs}$ data (using the
transfer function) in the Kerguelen area (Indian sector of the Southern Ocean). We revealed good
convergence of $JO_2$ rates from these two approaches, further supporting the Dehairs's function to
estimate POC remineralization rates in different biogeochemical settings of the Southern Ocean.
Here, we further investigate relationships between the mesopelagic $Ba_{xs}$ proxy, prokaryotic
activity and oxygen dynamics (Figure 1a) in the northwestern Mediterreanean Sea (MedSea), a
different biogeochemical setting to those already studied (see references above). Today,
observations of the various components of the MedSea biological C pump provide organic C
fluxes varying by at least an order of magnitude [Santinelli et al., 2010; Ramondenc et al., 2016].



Malanotte-Rissoli et al. [2014] reviewing unsolved issues and future directions for MedSea
research highlighted the need to further investigate biogeochemical processes at intermediate
(mesopelagic) and deep layers to reconciliate the C budget in the Mediterranean basin. Previous
particulate $Ba_{xs}$ dataset is very scarce in the NW- MedSea, with in general very low vertical
sampling resolution [Sanchez Vidal et al., 2005] or very restricted studied areas [Dehairs et al.,
1987; Sternberg et al., 2008]. Here we discuss $Ba_{xs}$, PHP and $JO_2$ (from optodes measurement
during incubations) at the ANTARES / EMSO-LO observatory site (Figure 1a, b). We
hypothesize that the Dehairs's transfer function converting $Ba_{xs}$ into POC remineralization also
applies in a different ocean ecosystem functioning from the Southern Ocean. We suggest that the
$Ba_{xs}$ proxy can be used as routine tracer to estimate local-scale processes of mesopelagic POC
remineralization in the Mediterranean basin.

**2. SAMPLING AND ANALYSES**
**2.1 STUDY SITE**
The BATMAN cruise (https://doi.org/10.17600/16011100, March 10-16 2016, *R/V* EUROPE)
took place to the ANTARES / EMSO-LO observatory site (42°48'N, 6°10'E; Tamburini et al.,
2013), 40 km off the coast of Toulon, southern France (Figure 1b). The hydrological and
biogeochemical conditions at this site are monitored monthly in the framework of the MOOSE
(Mediterranean Ocean Observing System for the Environment) program and of the EMSO
(European Multidisciplinary Subsea Observatory) observation program. The hydrography
displays the general three-layer MedSea system with surface, intermediate and deep waters
[Hainbucher et al., 2014]. Briefly, the main water masses can be distinguished (see potential
temperature – salinity diagram during the BATMAN cruise in Figure 1c): (1) Surface Water
(SW); (2) Winter Intermediate Water (WIW) and Levantine Intermediate water (LIW). LIW is





present at intermediate depths (around 400 m at ANTARES) and is characterized by temperature
and a salinity maxima; (4) Mediterranean Deep Water (MDW).

**2.2 ANALYSES**
For particulate barium, 4 to 7 L of seawater sampled using Niskin bottles were filtered onto 47
mm polycarbonate membranes (0.4 μm porosity) under slight overpressure supplied by filtered
air. Filters were rinsed with few mL of Milli-Q grade water to remove sea salt, dried (50°C) and
stored in Petri dishes. Thirteen depths between surface and 2000 m were sampled by combining
different casts sampled closeby in time and space (total of 28 samples). In the laboratory, we
performed a total digestion of filters using a tri-acid (0.5 mL HF /1.5 mL $HNO_3$ / HCl 1 mL; all
Optima grade) mixture in closed teflon beakers overnight at 95°C in a clean pressurized room.
After evaporation close to dryness, samples were re-dissolved into 10 mL of $HNO_3$ 2%. The
solutions were analysed for Ba and other elements of interest (Na and Al) by HR-ICP-MS (High
Resolution-Inductively Coupled Plasma- Mass Spectrometry; ELEMENT XR ThermoFisher).
Details on sample processing and analysis are given in Cardinal et al. [2001] and Jacquet et al.
[2015]. The presence of sea-salt was checked by analysing Na and the sea-salt particulate Ba
contribution was found negligible. Particulate biogenic barium in excess (hereafter referred to as
$Ba_{xs}$) was calculated as the difference between total Ba and lithogenic Ba using Al as the
lithogenic reference element [Taylor and Mc.Lennan, 1985]. The standard uncertainty [Ellison et
al., 2000] on $Ba_{xs}$ concentration ranges between 5.0 and 5.5%. The term "in excess" is used to
indicate that concentrations are larger than the $Ba_{xs}$ background. The background (or residual
value) is considered as "preformed" $Ba_{xs}$ at zero oxygen consumption left over after transfer and
partial dissolution of $Ba_{xs}$ produced during degradation of previous phytoplankton growth events.



131 Oxygen concentrations were measured using optical oxygen sensor (Aanderaa 4330-Optodes)

132 at 4 depths in the mesopelagic layer (175, 250, 450 and 1000 m). In total each of the 8 optodes

133 (two per depths) were placed into a sealed 1L borosilicate glass bottles incubated at a fixed

134 temperature of 13°C in thermo-regulated baths for 24 to 48 hours. Oxygen consumption rates

135 (later referred to as $JO_2$-Opt) were calculated from oxygen concentration evolution with time

136 applying linear model calculations.

137 Prokaryotic heterotrophic production (PHP) estimation was measured over time course

138 experiments at *in situ* temperature (13°C) following the protocol described in Tamburini et al.

139 [2002]. [3]H-leucine labelled tracer [Kirchman, 1993] was used. To calculate prokaryotic

140 heterotrophic production, we used the empirical conversion factor of 1.55 ng C per pmol of

141 incorporated leucine according to Simon and Azam [1989], assuming that isotope dilution was

142 negligible under these saturating concentrations.

143

## 3. RESULTS AND DISCUSSION

### 3.1 Barium vertical distribution

146 Particulate biogenic $Ba_{xs}$, particulate Al (pAl) and biogenic Ba fraction profiles in the upper

147 1000 m at ANTARES are reported in Figure 2a. $Ba_{xs}$ concentrations range from 12 to 719 pM.

148 The biogenic Ba fraction range from 51 to 91 % of the total particulate Ba signal. Particulate Al

149 concentrations (pAl) are low and range from 8 to 170 nM. $Ba_{xs}$ concentrations are low in surface

150 water (<100 pM) where the lithogenic fraction reaches 43 to 49 % in the upper 70 m. From

151 previous studies we know that $Ba_{xs}$ in surface waters is distributed over different, mainly non-

152 barite biogenic phases, and incorporated into or adsorbed onto phytoplankton material. As such

153 these do not reflect POC remineralization processes, in contrast to mesopelagic waters where

154 $Ba_{xs}$ is mainly composed of barite formed during prokaryotic degradation of organic matter. At



ANTARES the $Ba_{xs}$ profile displays a mesopelagic $Ba_{xs}$ maximum between 100 and 500 m,
reaching up to 719 pM at 175 m. Ba is mostly biogenic at these depths (> 80 %). $Ba_{xs}$
concentrations then decrease below 500 m to reach a background value of around 130 pM (see
BKG in Figure 2). Note that the MedSea is largely undersaturated with respect to barite, with
saturation state ranging between 0.2 and 0.6 over the basin [Jacquet et al., 2016; Jullion et al.,
2017]. For comparison, the Ba background value in the Southern Ocean reaches 180 to 200 pM
below 1000 m [Dehairs et al., 1997; Jacquet et al. 2015]. Previously, Sternberg et al. [2008]
reported the seasonal evolution of $Ba_{xs}$ profiles at the DYFAMED station (43°25'N-7°52'E;
BARMED project) northeast from ANTARES (Figure 1c) in the NW-MedSea. The present $Ba_{xs}$
profile at ANTARES (March 2016) is very similar to the $Ba_{xs}$ profile measured in March 2003 at
DYFAMED (Figure 2a). The slight difference between $Ba_{xs}$ profiles in the upper 75 m suggests
more Ba bounded and/or adsorbed onto phytoplankton material during BARMED. Both profiles
present a $Ba_{xs}$ maximum in the upper mesopelagic zone between 150 and 200 m. Below this
maximum, $Ba_{xs}$ concentrations gradually decrease to reach around 130 pM between 500 and
1000 m (this study). A similar value was reached between 500 and 600 m at the DYFAMED
station over the whole studied period (between February and June 2003; Sternberg et al., 2008).

**3.2 Prokaryotic heterotrophic production**
The particulate excess Ba (>BKG) is centred in the upper mesopelagic zone between 100 and
500 m and reflects that POC remineralization mainly occurred at these depths (Figure 2a). Depth-
weighted average (DWA) $Ba_{xs}$ content (409 pM) was calculated over this entire depth interval.
Figure 2b shows column-integrated PHP at 100 m over column-integrated PHP at 500 m
(PHP100/500= 0.90), according to the relationship obtained during KEOPS1 (summer) and
KEOPS2 (spring; out plateau stations) cruises in the Southern Ocean [Jacquet et al., 2008; 2015]

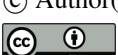



and #DY032 cruise (2015, *R/V* DISCOVERY) at the PAP (Porcupine Abyssal Plain) observatory
in the northeast Atlantic (49°N, 16.5 °W) (personal data). Results at the ANTARES / EMSO-LO
site follow the trend previously reported in the Southern Ocean, indicating higher DWA $Ba_{xs}$ in
situations where a significant part of column-integrated PHP is located deeper in the water
column (high Int. PHPx1/IntPHPx2 ratio; Figure 2b). These previous studies revealed that the
shape of the column-integrated PHP profile (i.e. the attenuation gradient) is important in setting
the $Ba_{xs}$ signal in the mesopelagic zone (Dehairs et al., 2008; Jacquet et al., 2008, 2015]. Indeed,
mesopelagic $Ba_{xs}$ appears reduced when most of the column-integrated PHP is limited to the
upper layer (indicating an efficient remineralization in surface), compared to situations where a
significant part of integrated PHP is located deeper in the water column (reflecting significant
deep PHP activity, POC export and subsequent remineralization) (Figure 2b). Our MedSea
results are located along the trend defined in the Southern Ocean during KEOPS1 cruise. It is
generally considered that $Ba_{xs}$ (barite) forms inside sulfate and/or barium oversaturated biogenic
micro-environments during POC degradation by heterotrophic prokaryotes. However, it is
unclear whether barite formation at mesopelagic depths is (directly or indirectly) bacterially
induced or bacterially influenced. Overall, our results strengthen the close link between the water
column $Ba_{xs}$ distribution and respiration (organic matter degradation).

**3.3 Oxygen- barium relationship**
The relationship we obtained at ANTARES between $Ba_{xs}$ concentrations and oxygen
consumption rates from optodes measurements (JO$_2$-Opt) is reported in Figure 3a. JO$_2$-Opt range
from 0.11 to 5.85 $\mu$mol L$^{-1}$ d$^{-1}$. The relationship indicates higher $Ba_{xs}$ concentrations with
increasing JO$_2$-Opt. An interesting feature is the intercept at zero JO$_2$-Opt (around 128 pM)





which further supports the Ba BKG value at ANTARES (130 pM) determined from measured
$Ba_{xs}$ profiles (Figure 3a).
We applied a similar approach as reported in Jacquet et al. (2015) where we show the
correlation between $JO_2$ obtained from dark community respiration DCR (winkler titration; $JO_2$-
DCR) data integration in the water column and $JO_2$ based on $Ba_{xs}$ content (Dehairs's transfer
function; later referred to as $JO_2$-Ba). Similarly, to estimate $JO_2$-Ba in the present study we used
the following equation [Dehairs et al., 1997]:

$JO_2$-Ba= $(Ba_{xs} - Ba\ BKG)/17450$     (1)

A Ba BKG value of 130 pM was used (see above). $JO_2$-Ba is confronted to $JO_2$-Opt integrated
over the same layer depth (between 175 and 450 m; Figure 3b). $JO_2$ rates are of the same order of
magnitude ($JO_2$-Ba= 4.59 mmol m$^{-2}$ d$^{-1}$ and $JO_2$-opt= 3.14 mmol m$^{-2}$ d$^{-1}$). The slight difference
could be explained by the integration time of both methods: few hours to days for the incubations
vs. few days to weeks for $Ba_{xs}$ (seasonal build-up; Jacquet et al., 2007). $JO_2$ rates calculated in the
present work are 3 times higher than those reported in the Southern Ocean during KEOPS1
[Jacquet et al., 2015] but they are in good agreement with the $Ba_{xs}$ vs $JO_2$ trend (Figure 3b).
Overall, our results indicate similar $Ba_{xs}$ - $JO_2$ relationship in the Southern Ocean and the
Mediterranean Sea. This further supports the universal validity of the Dehairs's transfer function
in the present study.

**3.4 Estimated particles remineralisation rates and implications**

In order to provide a $Ba_{xs}$-derived estimate of POC remineralization rate (MR) at the
ANTARES / EMSO-LO observatory during BATMAN cruise, we converted $JO_2$-Ba into C
respired using the Redfield (RR) C/$O_2$ molar ratio (127/175; Broecker et al., 1985) multiplied by
the depth layer considered (Z) [Dehairs et al., 1997]:



$MR = Z \times JO_2 - Ba \times RR$   (2)
We obtain a POC remineralization rate of 11 mmol C $m^{-2}$ $d^{-1}$ (10% RSD). This is within the
range of previously published remineralization fluxes in the Mediterranean Sea from sediment
trap [Sanchez-Vidal et al., 2005] and from thorium-derived data [Speicher et al., 2006]. It is also
in good agreement with recent POC flux attenuation combining drifting sediment traps and
underwater vision profilers [Ramondenc et al., 2016].
The present paper brings a first insight into the connections of $Ba_{xs}$, PHP and $JO_2$ at the
ANTARES/EMSO-LO observatory site in the northwestern Mediterranean Sea during the
BATMAN (2016) cruise. Our results reveal a strong relationship between $Ba_{xs}$ contents and
measured $JO_2$ rates. Also, DWA $Ba_{xs}$ vs. column integrated PHP, as well as measured vs. $Ba_{xs}$-
based $JO_2$ relationships follow trends previously reported in the Southern Ocean where the
Dehairs's function was first established to estimate POC remineralisation rate. Results from the
present study would indicate that this function can also be applied in the Mediterranean basin
provided that adequate $Ba_{xs}$ background values are estimated. From a global climate perspective,
the $Ba_{xs}$ tool will help to better balance the MedSea water column C budget. It will contribute to
gain focus on the emerging picture of the C transfer efficiency (strength of the biological pump).

**ACKNOWLEDGEMENTS**
We thank the officers and crew of *R/V* EUROPE for their assistance during work at sea. This
research was supported by the French national LEFE/INSU "REPAP" project (PI. S. Jacquet). It
was co-funded by the "ROBIN" project (PIs. C. Tamburini, F.A.C. Le Moigne) of Labex OT-
Med (ANR-11-LABEX-0061) funded by the Investissements d'Avenir and the French
Government project of the ANR, through the A*Midex project (ANR-11-IDEX-0001-02).
Authors have benefited of the support of the SNO-MOOSE and SAM-MIO. BATMAN is a





contribution to the "AT – POMPE BIOLOGIQUE" of the Mediterranean Institute of
Oceanography (MIO) and to the international IMBER program. The instrument (ELEMENT XR,
ThermoFisher) was supported in 2012 by European Regional Development Fund (ERDF).





**Figure captions**
Figure 1: (a) Schematic representation of the convergence of the different estimators of oxygen
consumption and C remineralization rates from the "oxygen dynamics", "barium proxy" and
"prokaryotic activity" tools; (b) Location of the BATMAN cruise at the ANTARES observatory
site in the NW-Mediterranean Sea (42°48'N, 6°10'E); (b) Potential temperature - salinity - depth
plots and isopycals for BATMAN profiles. SW : Surface Water, WIW : Winter Intermediate
Water, LIW : Levantine Intermediate Water, DMW : Deep Mediterranean Water. Graph
constructed using Ocean Data View (Schlitzer, 2002; Ocean Data View; http://www.awi-
bremerhaven.de/GEO/ODV)

Figure 2: (a) Particulate biogenic $Ba_{xs}$ (pM) and particulate Al (nM) profiles next to the biogenic
Ba fraction (%) in the upper 1000 m at ANTARES. The grey area represents a biogenic Ba
fraction larger than 80 %. BKG: $Ba_{xs}$ background. $Ba_{xs}$ profile (pM) at DYFAMED : data  from
Sternberg et al. (2008); (b) ANTARES ratio plot (green square) of integrated PHP in the upper
100 m over integrated PHP in the upper 500 m versus depth-weighted average (DWA)
mesopelagic $Ba_{xs}$ (pM) over the 150- 500 depth interval. Regression of the same ratio is reported
for KEOPS1 (out plateau stations) and KEOPS2 (Southern Ocean; Jacquet et al., 2015) and
#DY032 (PAP station, NE-Atlantic; pers. data) cruises.

Figure 3: (a) Relationship between $Ba_{xs}$ concentrations (pM) and oxygen consumption rates
($\mu mol\ L^{-1}\ d^{-1}$) from optodes measurements (JO$_2$-Opt) at ANTARES; (b) Confrontation of oxygen
consumption rates (mmol m$^{-2}$ d$^{-1}$) obtained from different methods: optodes measurements (this
work), dark community respiration DCR (winkler titration; JO$_2$-DCR; Jacquet et al., 2015;



KEOPS1), and (Dehairs's transfer function calculation based on $Ba_{xs}$ content (Dehairs et al.,

1997).






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





Figure 1

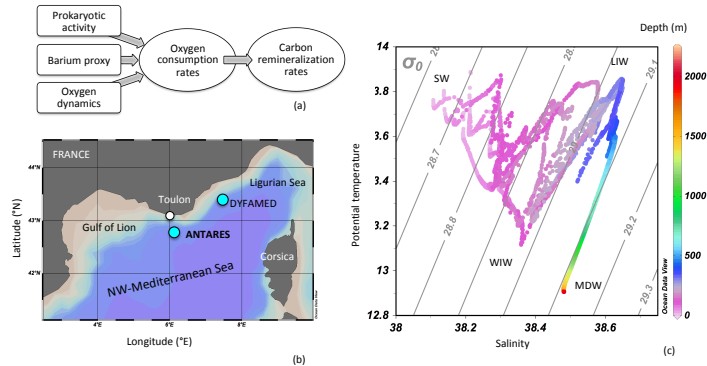




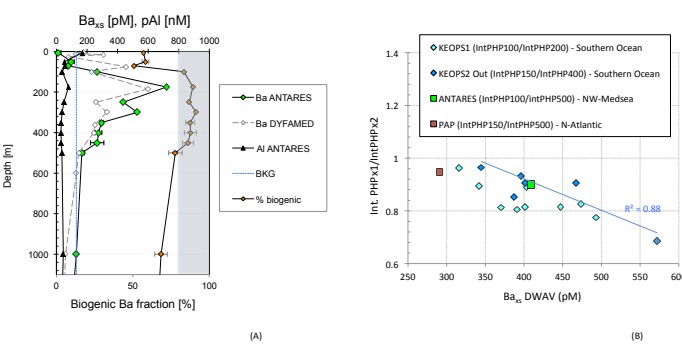

Figure 2





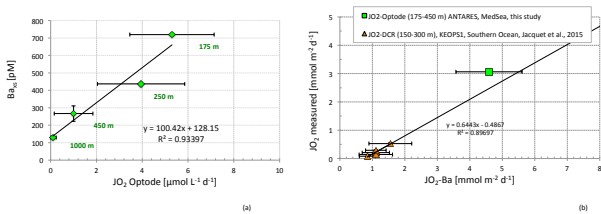

Figure 3