# Peer review of "On the barium - oxygen consumption relationship in the Mediterranean Sea: implications"

_Biogeosciences, 2020_

## Referee Comment (RC1) · Anonymous Referee #1 · 15 Sep 2020

Jacquet et al. present new data of Baxs concentrations, O2 consumption rates from direct measurements and prokaryotic heterotrophic productions (PHP) from the ANTARES station in the Mediterranean Sea. The aim of this research is to investigate the connections between these three parameters (Baxs concentrations, O2 consumption rates and PHP) in order to validate the Baxs-O2 consumption transfer function first proposed by Dehairs et al. (1997) in the Southern Ocean. The authors found higher Baxs concentration associated to deeper PHP and to greater O2 consumption rate. Finally, they found a relatively good agreement between O2 consumption rates estimated by the Baxs transfer function from the Southern Ocean (Dehairs et al., 1997) and by direct measurements, confirming the use of this transfer function in the

[Figure]

Mediterranean Sea. Previous studies used Baxs as a tracer of O2 consumption and thus as a tracer of POC remineralisation, but they either assumed the universality of the Southern Ocean transfer function (e.g. Cardinal et al., 2005) or proposed new transfer function without direct O2 consumption measurements (e.g. Lemaitre et al., 2018). It is therefore of interest to investigate the conformity of this transfer function by directly measuring O2 consumption rates and PHP. For that reason, the findings of this study are highly valuable for the community. However, the authors report data from only one station (only one data added in both the PHP/Baxs and JO2-Baxs/JO2-opt correlations) which is weak to support their conclusions. Statistical analyses (p-values, errors on the slopes, etc) are needed. Also, a direct comparison of the Baxs/JO2-opt correlation from this study (where the authors show 4 data points; Fig. 3a) with the one proposed by Dehairs et al. (1997) in the Southern Ocean would be very useful and more convincing, to me. Many details are also missing in the methods to really understand how Baxs concentrations, O2 consumption rates and PHP were measured. Moreover, I would appreciate if there was a discussion about the variations found between ANTARES, PAP and DYFAMED stations, about the differences observed between the Southern Ocean and Mediterranean Sea correlations (Baxs background for example) and about the implications of this study in the water column C budget of the Mediterranean Sea. Finally, all the data (Baxs concentrations, O2 consumption rates and PHP) should be presented in a Table. Please, see all my comments in the attached file. Overall, the manuscript is well written and will be a good fit for publication in Biogeosciences, but considering the lack of details and comparisons, considering the relatively large error bar associated to the JO2-opt, and considering that this study adds only one data point to the JO2 correlation, I would suggest the authors to soften their conclusion on the 'universal validity' of the Dehairs's transfer function.

Please also note the supplement to this comment:
https://bg.copernicus.org/preprints/bg-2020-241/bg-2020-241-RC1-supplement.pdf

[Figure]

**Supplement:**

Jacquet et al. present new data of $Ba_{xs}$ concentrations, $O_2$ consumption rates from direct measurements and prokaryotic heterotrophic productions (PHP) from the ANTARES station in the Mediterranean Sea. The aim of this research is to investigate the connections between these three parameters ($Ba_{xs}$ concentrations, $O_2$ consumption rates and PHP) in order to validate the $Ba_{xs}$-$O_2$ consumption transfer function first proposed by Dehairs et al. (1997) in the Southern Ocean. The authors found higher $Ba_{xs}$ concentration associated to deeper PHP and to greater $O_2$ consumption rate. Finally, they found a relatively good agreement between $O_2$ consumption rates estimated by the $Ba_{xs}$ transfer function from the Southern Ocean (Dehairs et al., 1997) and by direct measurements, confirming the use of this transfer function in the Mediterranean Sea.

Previous studies used $Ba_{xs}$ as a tracer of $O_2$ consumption and thus as a tracer of POC remineralisation, but they either assumed the universality of the Southern Ocean transfer function (e.g. Cardinal et al., 2005) or proposed new transfer function without direct $O_2$ consumption measurements (e.g. Lemaitre et al., 2018). It is therefore of interest to investigate the conformity of this transfer function by directly measuring $O_2$ consumption rates and PHP. For that reason, the findings of this study are highly valuable for the community.

However, the authors report data from only one station (only one data added in both the PHP/$Ba_{xs}$ and $JO_2$-$Ba_{xs}$/$JO_2$-opt correlations) which is weak to support their conclusions. Statistical analyses (p-values, errors on the slopes, etc) are needed. Also, a direct comparison of the $Ba_{xs}$/$JO_2$-opt correlation from this study (where the authors show 4 data points; Fig. 3a) with the one proposed by Dehairs et al. (1997) in the Southern Ocean would be very useful and more convincing, to me. Many details are also missing in the methods to really understand how $Ba_{xs}$ concentrations, $O_2$ consumption rates and PHP were measured. Moreover, I would appreciate if there was a discussion about the variations found between ANTARES, PAP and DYFAMED stations, about the differences observed between the Southern Ocean and Mediterranean Sea correlations (Baxs background for example) and about the implications of this study in the water column C budget of the Mediterranean Sea. Finally, all the data ($Ba_{xs}$ concentrations, $O_2$ consumption rates and PHP) should be presented in a Table.

Overall, the manuscript is well written and will be a good fit for publication in Biogeosciences, but considering the lack of details and comparisons, considering the relatively large error bar associated to the $JO_2$-opt, and considering that this study adds only one data point to the $JO_2$ correlation, I would suggest the authors to soften their conclusion on the 'universal validity' of the Dehairs's transfer function.

**A. Specific comments**

**1-Introduction**

In general, this section should be developed and should mention all the studies on Baxs concentrations as a tracer of POC remineralisation but also all the recent studies investigating barite formation and the role of barite on the Ba cycle.

Lines 55-61: Please develop the paragraph about the use of Baxs as a geochemical proxy: more studies have worked on Baxs in the past (e.g. Bishop, 1988; Collier and Edmond, 1984; Ganeshram et al., 2003; Gonzalez-Munos et al., 2003). Moreover, there are some recent studies that should be mentioned/discussed about lab experiments and Ba isotopes giving extremely interesting new insights on the formation of barites and on their role in the Ba cycle in the ocean. Please, see for example the studies of Martinez-Ruiz et al. (2018, 2019), Horner et al. (2015), Cao et al. (2020), Hsieh et al. (2017).

Lines 66-68: Lemaitre et al. (2018) do not use the Southern Ocean transfer function proposed by Dehairs et al. (1997). These authors proposed a new function specific to the North Atlantic. Consequently, they (sort of) 'revised' the validity of this transfer function at least for the GEOVIDE study area. You could also use this study as an additional reason to check the validity of the Southern Ocean transfer function in the Mediterranean Sea.

Line 67: Instead of Lemaitre et al. (2018), you could also cite Cardinal et al. (2005), Dehairs et al. (2008), Jacquet et al. (2008 a, b, 2011, 2015), Planchon et al. (2013).

**2-Sampling and Analyses**

This section must be developed. The reader needs more information and details on how and how well you measure Baxs, $O_2$ consumption rates from optodes and prokaryotic heterotrophic productions. Also, please show all your new data in Figures and Tables.

Line 115: Please show the full Baxs, pAl and bio Ba depth profile, i.e. from surface to 2000m, in Fig.2a. This will also confirm that the Baxs background stays at around 130pM at depths > 500m.

Lines 115-116: If I am correct, the samples used for the data presented in Fig. 2a have not been collected on the same day or exact location. Please prove that there was no evolution of water mass or biology between each sampling. If there was any change, could this influence your Baxs or pAl concentrations?

Lines 120-121: Please give the precision and accuracy of your analyses.

Lines 124-125: 'sea-salt particulate Ba contribution was found negligible'. What is negligible? Please give numbers.

Line 125: Give more details on the Ba/Al ratio you are using to correct the lithogenic fraction. I suppose it is from the UCC but how does this value compare to the lithogenic inputs at ANTARES? This station is relatively close to the coast and is likely subject to lithogenic inputs, it is therefore important to be sure about the Ba/Al ratio used to correct the lithogenic fraction. Without that, your estimation of Baxs concentrations may not be correct. For comparison, Lemaitre et al. (2018) do not take into account data from two stations where the pBa-litho accounts for 28 and 44% of total Ba. At ANTARES, the Ba biogenic fraction range from 50 to 80%, meaning the lithogenic fraction is not negligible.

Line 126: How did you determine the standard uncertainty? From the RSD given by the Element for Ba? From error propagation, taking into account the RSD of Ba and Al?

Lines 131-136: There is no reference at all in this paragraph – it is thus difficult to understand the technic for someone who is not familiar with this. Please explain, at least briefly, how you measure O2 concentrations with this technic and how you calculate the O2 consumption rates – an equation might help? Can you prove the precision/accuracy of this method? I suppose you need relatively precise measurements to determine an O2 consumption rate. However the errors associated to this measurement and to the final calculation seem high (Fig. 3), why?

Lines 137-142: Same here, please give more details on the protocol you use for determining the PHP. Why do you use 3H-leucine? How do you then calculate the PHP (equation)?

**3-Results and Discussion**

The authors should give more details to convince the reader about the validity of this Baxs-JO2 function in the MedSea. A direct comparison of the slope of the transfer function you obtain here (Fig. 3a) with the one from the Southern Ocean would be helpful. Some statistics would also help. Moreover, I think this section would get more interesting if there was some comparison with the literature and some explanations on why some of your results slightly differ compared to those of other study areas (essentially, more explanation on the story of the MedSea data – not only about the use of Baxs in this area to trace O2 consumption). The figures could be clearer as well.

Line 149: 'pAl concentrations are low...' 170nM is not low! On the contrary, it clearly shows a lithogenic input and this makes your Baxs estimations doubtful as the lithogenic correction may not be perfectly constrained. How much is the pBa lithogenic fraction in the depth layer that is interesting for this study (i.e. 100-500m)? Can it be considered as negligible? If yes, why? Please see my previous comment about the Ba/Al ratio and discuss more about the lithogenic correction at ANTARES station.

Line 156: You mention the pBa biogenic fraction in the interested depth layer is >80% but is it high enough to be assured of a good Baxs estimation? What is the error associated to this correction (this could go to the methods section)?

Lines 157-160: How do you explain the difference between the Baxs background observed in the Southern Ocean and in your study? For example, Lemaitre et al. (2018) also observed a Baxs background at 180pM in the North Atlantic. What is different in the MedSea?

Lines 165-166: How do you explain the difference of adsorption between ANTARES and DYFAMED stations? Is it related to different bloom timing or intensity?

Lines 168-169: Please show the full depth profile, i.e. from the surface to 2000m, in Fig. 2a. That would be useful to clearly see the background level.

Lines 169-170: At DYFAMED station, Baxs concentrations seem to keep decreasing for depths >600m, why is it not stabilised at 130pM?

Lines 180-183: Please, discuss the result of the PAP station if you present it. It is below the trend, why? Moreover, what is the p-value of this trend? Is it a significant correlation with and without the new ANTARES and PAP data? Is it possible to add data from DYFAMED?

Line 184: Are these PHP profiles similar to the one at ANTARES station? Could you plot them all in a figure and add the ANTARES data in a table?

Lines 185-189: 'Indeed, mesopelagic Baxs...' These lines repeat your sentence lines 181-183 '..indicating higher DWA Baxs in situations where a significant part..'. Please re organise this section to avoid repeating things.

Lines 189-190: 'Our MedSea resultS are located..'. You provide only one new result from ANTARES station, please change the plural to singular form in this sentence. Also, this sentence repeats what you say lines 180-181. Maybe you should delete it.

Lines 190-195: Please develop this section according to the new literature (e.g., Martinez-Ruiz et al., Horner et al., Cao et al., Hsieh et al..) and find a transition with your previous sentence.

Line 200, Fig. 3a and b: It seems that there is a mistake with the units. They do not correspond to those in Jacquet et al. (2015), would it not be mmol/m2/d instead? If I am correct, please change all your JO2 data in umol/L/d and compare the slope you obtain in Fig. 3a with the one from the Dehairs et al. (1997).

Lines 201-203: I agree this is a very interesting feature confirming your background Baxs concentration! Could this result give an insight on why there is a different Baxs background in the MedSea compared to other areas?

Line 209: Why do you use a factor of 17450 here while it is 17200 in Jacquet et al. (2008) or Lemaitre et al. (2018)?

Lines 214-219: There is a large error bar associated to your ANTARES data point for JO2-opt (Fig. 3b), why? I agree that considering this large error bar, your data fits the trend observed during KEOPS. However, this large error bar and the poor distributions of the data points (either low JO2 for KEOPS or high JO2 for ANTARES) make this correlation too weak to state that there is no difference between both regions. What is the p-value of this correlation with and without ANTARES? Is it possible to add data from PAP or DYFAMED stations? I would be more convinced by a comparison of your Baxs-JO2 trend with the one of the Southern Ocean. For now, the slope in Fig. 3a is very different from the one of the Southern Ocean (100 versus 17450). After fixing the unit problem, please discuss about this comparison.

Line 226: Please indicate what is Z in this study.

Lines 228-229: Please give the range of the fluxes from the literature and discuss them according to the one you estimate at ANTARES.

Lines 239-241: Expand a bit the discussion here. How does your study contribute to the MedSea carbon budget? Does it help balancing the water column budget?

B. **Line notes**

Abstract:
Lines 25-27: These are not new observations/conclusions. Please make it clear here that you are confirming what has been observed earlier in another area (Jacquet et al., 2015).

Line 25: 'higher Baxs (409 pM; 100- 500 m) [occurs] in situations where integrated PHP (PHP100/500= 0.90) is located deeper'

Line 26: 'higher Baxs [occurs] with increasing JO2-Opt'

Introduction:
Line 63: 'highly resolved, precise..' seems a bit exaggerated as a sampling resolution of 50m depth is good but not high for me and I suppose the technics may be more precise today compared to 1997.

Line 70: I would delete this sentence as it repeats the sentence line 68 ('Yet its validity has never been tested..') and it separates two linked sentences.

Line 70: 'These advancements..' refer to the results of Jacquet et al. (2015) I suppose? Please make it clearer.

Line 83: Which fluxes are you referring here? Primary production, export, remineralistion?

Line 83: Please give a range of the fluxes determined by Santinelli et al (2010) and Ramondec et al. (2016).

Methods:
Line 111: I would name this whole section 'Methods' and would name the sub-section 2.2 'Sampling and Analyses'

Line 107: 'and [(3)] Levantine Intermediate Water…'

Line 117: 'total digestion of filters using a [concentrated] tri-acid mixture..'

Line 130: 'The background (or residual value) is considered as "preformed" Baxs at zero oxygen consumption left over after transfer and partial dissolution of Baxs produced during degradation of previous phytoplankton growth events. [The background is set at 130pM in this study].'

Results and Discussion:
Line 145: Maybe modify to '[Particulate Baxs] vertical disctribution' to avoid any confusion for the reader.

Line 160: 'For comparison, the [Baxs] background value…'

Line 173: 'the particulate excess Ba (>BKG)' is confusing for me. You never expressed Baxs like this before. Please keep the same wording all along the manuscript, maybe modify to 'The maxima Baxs concentrations are centred..'

Line 174: 'in this depth layer' instead of 'at these depths'

Lines 174-175: Explain what is the depth-weighted average, as you did for example in Jacquet et al. (2015): 'i.e. the Baxs inventory divided by the depth layer considered Z'.

Line 175: 'over the 100-500m depth layer' instead of 'this entire depth layer'. It will avoid any confusion with Fig.2b and all the different depth integrations.

Line 176: 'Figure 2b shows [the] column-integrated PHP at 100m over the [one] at 500m (PHP100/500). Our PHP100/500 ratio at ANTARES station is of 0.90 and is compared to results obtained during KEOPS1…'

Line 180: 'ResultS at the ANTARES..' Are there more than one result? On Fig.2b, there is only one data from ANTARES station.

Line 181: '…follow the trend previously reported in the Southern Ocean [(blue dashed line in Fig.2b; Jacquet et al., 2015)]..'

Lines 181-182: Please make it clear that the ANTARES data confirms the conclusions found in Jacquet et al. (2015) and that it is not a new conclusion.

Line 204: '[In Figure 3b,] we applied..'

Line 217: 'Overall, our results indicate [a] similar Baxs-JO2 relationship..'

Lines 257-258: You also show the DYFAMED station in this figure. Please mention it is for comparison and cite Sternberg et al. (2008).

Line 258: '[c] potential temperature-salinity-depth plots…'

Line 269: Could you integrate the DWA Baxs between 100-500m as well (to match with the PHP integration)?

Lines 269-271: 'Regression of the same ratio is reported for KEOPS1 ([light blue symbols;] out plateau stations) and KEOPS2 ([dark blue symbols;] Southern Ocean; Jacquet et al., 2015) and #DY032 ([red square;] PAP station, NE-Atlantic; pers. data) cruises.'

Lines 269-271: Please clarify what the blue dashed line represents. Is it from Jacquet et al. (2015) or does it take into account all data points including the new ones from ANTARES and PAP stations?

Line 275: mmol/m2/d instead?

Lines 275-276: '..optode measurements (this study; [green square]), dark community respiration DCR (winkler titration; [red triangles]; JO2-DCR; Jaquet et al., 2015; KEOPS1)'

Lines 277-278: It is not clear if you speak about the y-axis or the black line. I propose to re write as '…and [Baxs contents (Southern Ocean transfer function from Dehairs et al. (1997); JO2-Ba]. The black line corresponds to the correlation found in Jacquet et al. (2015)'. If this is correct, please also mention that this correlation excludes some data points from A3 and E stations.

Figure 2a, in the legend: Ba[xs] ANTARES; Ba[xs] DYFAMED; [p]Al ANTARES. And please show the full depth profile (until 2000m).

Figure 2b: Please indicate from where the blue line comes from. And indicate the p-value.

Figure 3a: Please check the units and indicate JO2 in umol/L/d. And show the trend from Dehairs et al. (1997) in the Southern Ocean. Give the p-value.

Figure 3b: Please indicate JO2 in umol/L/d. Also, indicate from where the black line comes from. And indicate the p-value.

**C. References:**

**Bishop** J. K. B.: The barite-opal-organic carbon association in oceanic particulate matter, Nature, 332, 341–343, https://doi.org/10.1038/332341a0, **1988**

**Cardinal D**., Savoye N., Trull T. W., André L., Kopczynska E. E., and Dehairs F.: Variations of carbon remineralisation in the Southern Ocean illustrated by the Baxs proxy, Deep-Sea Res. Pt. I, 52, 355–370, https://doi.org/10.1016/j.dsr.2004.10.002, **2005**.

**Cao** Z., Li Y., Rao X., Yu Y., Hathorne E.C., Siebert C., Dai M., Frank M.: Constraining barium isotope fractionation in the upper water column of the South China Sea, Geochemica et Cosmochimica Acta, 288, 120-137, https://doi.org/10.1016/j.gca.2020.08.008, **2020.**

**Dehairs** F., Jacquet, S., Savoye, N., Van Mooy, B. a S., Buesseler, K. O., Bishop, J. K. B., Lamborg, C. H., Elskens, M., Baeyens, W., Boyd, P. W., Casciotti, K. L., Monnin, C.: Barium in twilight zone suspended matter as a potential proxy for particulate organic carbon remineralization: Results for the North Pacific, Deep-Sea Res. Pt. II, 55, 1673–1683, https://doi.org/10.1016/j.dsr2.2008.04.020, **2008**.

**Ganeshram** R.S., François R., Commeau J., Brown-Leger S.L.: An experimental investigation of barite formation in seawater, Geochim. Cosmochim. Ac., 67, 2599–2605, https://doi.org/10.1016/S0016-7037(03)00164-9, **2003**.

**Gonzalez-Munoz** M.T., Fernandez-Luque B., Martínez-Ruiz F., Chekroun K. Ben, Arias J.M., Rodríguez-Gallego M., Martínez-Canamero M., de Linares C., and Paytan A.: Precipitation of barite by Myxococcus xanthus: possible implications for the biogeochemical cycle of barium, Appl. Environ. Microbiol., 69, 5722–5725, https://doi.org/10.1128/AEM.69.9.5722-5725.2003, **2003**.

**Horner** T.J., Kinsley C.W., Nielsen S.G.: Barium-isotopic fractionation in seawater mediated by barite cycling and oceanic circulation, Earth and Planetary Science Letters, 430, 511-522, **2015**.

**Hsieh** Y.T., Henderson G.M.: Barium stable isotopes in the global ocean: Tracer of Ba inputs and utilization, Earth and Planetary Science Letters, 473, 269-278, **2017**.

**Jacquet** S.H.M., Savoye N., Dehairs F., Strass V.H., Cardinal D.: Mesopelagic carbon remineralization during the European Iron Fertilization Experiment, Global Biogeochem. Cy., 22, 1–9, https://doi.org/10.1029/2006GB002902, **2008a**.

**Jacquet** S.H.M., Dehairs F., Savoye N., Obernosterer I., Christaki U., Monnin C., Cardinal D.: Mesopelagic organic carbon remineralization in the Kerguelen Plateau region tracked by biogenic particulate Ba, Deep-Sea Res. Pt. II, 55, 868–879, https://doi.org/10.1016/j.dsr2.2007.12.038**, 2008b**.

**Jacquet** S.H.M., Dehairs F., Dumont I., Becquevort S., Cavagna A.-J., Cardinal, D.: Twilight zone organic carbon remineralization in the Polar Front Zone and Subantarctic Zone south of Tasmania, Deep-Sea Res. Pt. II, 58, 2222–2234, https://doi.org/10.1016/j.dsr2.2011.05.029, **2011**.

**Jacquet** S.H.M., Dehairs,F., Cavagna A. J., Planchon F., Monin L., André L., Closset I., Cardinal D.: Early season mesopelagic carbon remineralization and transfer efficiency in the naturally iron-fertilized Kerguelen area, Biogeosciences, 12, 1713–1731, https://doi.org/10.5194/bg-12-1713-2015, **2015**.

**Lemaitre** N., Planquette H., Planchon F., Sarthou G., Jacquet S., Garcia-Ibanez M.I., Gourain A., Cheize M., Monin L., André L., Laha P., Terryn H., Dehairs F., Particulate barium tracing of significant mesopelagic carbon remineralisation in the North Atlantic, Biogeosciences, 15, 2289-2307, https://doi.org/10.5194/bg-15-2289-2018, **2018.**

**Martinez-Ruiz** F., Jroundi F., Paytan A., Guerra-Tschuschke I., del Mar Abad M., Gonzalez-Munoz M. T.: Barium bioaccumulation by bacterial biofilms and implications for Ba cycling and use of Ba proxies, Nature Communications, 9, 1619, DOI: 10.1038/s41467-018-04069-z, **2018**.

**Martinez-Ruiz** F., Paytan A., Gonzalez-Munoz M.T., Jroundi F., del Mar Abad M., Lam P.J., Bishop J.K.B., Horner T.J., Morton P.L., Kastner M.: Barite formation in the ocean: Origin of amorphous and crystalline precipitates, Chemical Geology, 511, 441-451, https://doi.org/10.1016/j.chemgeo.2018.09.011, **2019**

**Planchon** F., Cavagna A.-J., Cardinal D., André L., Dehairs F.: Late summer particulate organic carbon export and twilight zone remineralisation in the Atlantic sector of the Southern Ocean, Biogeosciences, 10, 803–820, https://doi.org/10.5194/bg-10-803-2013, **2013**.

**Sternberg** E., Jeandel C., Robin E., Souhaut M.: Seasonal cycle of suspended barite in the Mediterranean Sea, Geochimica et Cosmochimica Acta, 72, 4020-4034, https://doi.org/10.1016/j.gca.2008.05.043, **2008**.

---

## Referee Comment (RC2) · Anonymous Referee #2 · 7 Oct 2020

The authors present new data concerning the relation between biogenic barium (Baxs), the O2 consumption and prokaryotic heterotrophic production (PHP) in the Mediterranean Sea. The purpose of this paper is to improve our understanding of the relation between barium and oxygen and to test the validity of the Dehairs transfer function in the Mediterranean Sea. This relation has never been tested in the Mediterranean Sea. They also investigated further the relation between PHP and Baxs distribution. I think the paper has nicely approached these issues with their new dataset. Although I think the dataset and the statistics of the study are weak and the paper is missing some important information. Nevertheless, such information is still valuable for the community and may help to improve our understanding of barium cycle in the ocean. I would

recommend the manuscript for publication in Biogeosciences. However, I list issues below, which I think the authors should consider in their revision:

My main concern for this paper is that the authors conclude that there is strong relationship between Baxs and JO2 rates and that the transfer function can be apply with no restriction in the MedSea. The authors should be more moderate about these statements considering that there are not that many data and the lack of statistical analysis for these relationships. Indeed, linear regressions in figures 2b), 3a) and b) should take into account the errors bars. The errors on the slope and intercept should be shown, as well as the p value to show if the relations are significant. On figure 2a), only data from KEOPS 2 are considered for the regression. The regression should take all the data (KEOPS 1; KEOPS 2 and PAP). Error bars of these data should be taking into account. Then, a 95% confidence interval could also be added to show that the ANTARES data point is in that interval. Concerning the JO2 from optode vs JO2 from Baxs (Figure 3a and the associated paragraph (lines 198–203)), the fact that the intercept matches the background is an interesting feature. However, this feature is biased by the fact that the regression is taking into account the value at 1000m (130pM). Indeed, this value from 1000m is used as the background and then use in the regression to prove that the background is close to 130pM. It is a circular reasoning. Indeed, this value (1000m≈30pM) forces the regression and so should not be used for that regression. The regression should take only value at 175m, 250m and 450m. The error bars for these values should also be taking into account in that regression. Errors on the slope and intercept should be provided especially if you are discussing the fact that the intercept match the background value. In this figure, it will also be interesting to see the data from the Southern Ocean (Dehairs et al., 1997) and the North Atlantic (Lemaitre et al., 2018) as a comparison. For the JO2 Ba vs JO2 measured relationship (figure 3b), the authors say that MedSea data are 3 times higher than KEOPS data. And there is only one point for the MedSea with important error bars. Considering all of that it seems hard to say that the MedSea show the same relationship than the Southern Ocean and even more saying that this support the universal validity of the Dehair's transfer

function. Maybe a 95% interval would be useful in this figure too. This interval would show that the ANTARES value is good agreement with the relationship from KEOPS data. More data would be needed to state the universal validity of the Dehair's transfer function.

Concerning the analyses part, different information is missing. First, only few information is provided on how pAl data have been generated. The authors should provide more information on the sampling, the analysis of these data and their accuracy. Moreover, the authors should elaborate why and how pAl used to correct Ba from the lithogenic fraction would help the reader. The authors do not provide any references for the measurement of the O2 consumption rates. More explanations and references are needed to help the reader understand how these data have been generated. Please also explain how from oxygen concentrations you obtain the consumption rates (linear model calculations), maybe with equation. Provide the accuracy of these data. In the same way, more information and references on PHP measurements and why PHP are interesting to compare to Ba and O2 (in the introduction) will make the rest of manuscripts easier to understand for the reader. Also the accuracy these data should be provided. In the manuscript and figures, different units are used the O2 consumption data, please verify and unify.

Finally, the data are never shown in tables, data should be presented in tables in the manuscript or at least in supplementary materials.

Specific comments:

Line 40: "In essence, the biological C pump is termed for the numerous processes involved in maintaining the vertical gradient in dissolved inorganic C." this sentence is not clear, please rephrase.

Lines 55–63: More references about the use of Baxs and recent studies about it will make this paragraph stronger.

Line 67: if this function has been "widely used", more than two references are expected, please add more references or change widely.

Line 68: "has never been tested in other oceanic provinces" is in contradiction with "widely used" and Lemaitre et al., 2018 used it in the North Atlantic. Please clarify.

Line 68 "significant progresses" what kind of progress? Please elaborate.

Line 69: "made in relating", not correct, it should be changed to "related to" or "In relation to"

Line 71: "related with" should be replaced by "related to".

Lines 71–73: how Baxs and PHP are related? What is the temporal progression of POC reminaralisation processes? Line 72: "the rate of change with depth" is not clear, please rephrase.

Line 75: "to derived JO2" should be replaced by "to JO2 derived"

Line 77: "convergence" is probably not the right word here maybe "agreement" would be better

Line 80: "Mediterreaneaan Sea" should be replaced by "Mediterranean Sea"

Lines 81-83: How the MedSea is different from the other studied regions? Please provide the fluxes from the cited references.

Line 84: "reviewing unsolved issues" should be rephrased

Line 116: "closeby" should be replaced by "close by"

Line 118: "Teflon" should be replaced by "Teflon"

Line 119: "After evaporation close to dryness" is not clear, does this mean that samples are almost dry? Please rephrase Line 124: "the sea-salt particulate Ba contribution was found negligible", what is the % of sea salt in samples? Line131-136: please provide more information and reference for this paragraph. Also explain the calculations

and/or show the equations to get the JO2-Opt. Line 137: "Over the time course exper-iments" should be "Over the time course of the experiments" Line 149: "pAl are low", please compare to literature data and reference for this statement.

Line 151: "distributed over different", one word is missing.

Line 154: "Baxs is mainly composed of barite formed during prokaryotic degradation of organic matter." Please add a reference for this statement.

Line 157: "background value of around 130 pM" Are values below 1000m constant? Why this value is lower than the southern ocean?

Line 160: "the Ba background" is this Ba or Baxs? Please make sure that all the occurrence of Ba and Baxs are the right one in all the manuscript.

Lines 165–166: How do you explain this differences in absorption between BARMED and ANTARES ?

Line 173: "centred" not clear, please rephrase

Lines 174–175: please elaborate the use of Depth-weight average

Line 177: "PHP100/500" this notation is different in figures and in the rest of the manuscript please unify all these notations

Lines 177–179: This sentence is not understandable, please rephrase

Line 209: Please justify why you choose 17450 in equation 1.

Line 210 : "confronted" should be replace by "compared"

Lines 240-241: How the Baxs will help to better balance the C budget in the MedSea ? Please elaborate that part.

Line 266 please precise the background value.

Line 267 : "(b) ANTARES ratio plot (green square) of integrated PHP in the upper 100 m over integrated PHP in the upper 500 m versus depth-weighted average (DWA)mesopelagic Baxs (pM) over the 150- 500 depth interval." is not clear it could be replace by "(b) ANTARES (green square) integrated PHP in the upper 100 m over integrated PHP in the upper 500 m versus depth-weighted average (DWA) mesopelagic Baxs (pM) over the 150- 500 depth interval"

Line 274: "confrontation" should be replaced by "comparison"

Line 277: unnecessary bracket.

Figures: Figure 1 c): The unit for missing potential temperature is missing. Some numbers are missing/hidden on the potential temperature axis.

Figure 2a): it would be great to see the profile deeper to see if the background stays constant below 1000m. pAl concentrations data don't have error bars, if not shown the errors should be described in the manuscript.
* * *

---

## Author Comment (AC1) · 1 Dec 2020

On the barium-oxygen consumption relationship in the Mediterranean Sea: implications for mesopelagic marine snow remineralization

Authors: Jacquet et al.

Response to Referee #1

Jacquet et al. present new data of Baxs concentrations, $O_2$ consumption rates from direct measurements and prokaryotic heterotrophic productions (PHP) from the ANTARES station in the Mediterranean Sea. The aim of this research is to investigate the connections between these three parameters (Baxs concentrations, $O_2$ consumption rates and PHP) in order to validate the Baxs-$O_2$ consumption transfer function first proposed by Dehairs et al. (1997) in the Southern Ocean. The authors found higher Baxs concentration associated to deeper PHP and to greater $O_2$ consumption rate. Finally, they found a relatively good agreement between $O_2$ consumption rates estimated by the $Ba_{xs}$ transfer function from the Southern Ocean (Dehairs et al., 1997) and by direct measurements, confirming the use of this transfer function in the Mediterranean Sea.

Previous studies used $Ba_{xs}$ as a tracer of $O_2$ consumption and thus as a tracer of POC remineralisation, but they either assumed the universality of the Southern Ocean transfer function (e.g. Cardinal et al., 2005) or proposed new transfer function without direct $O_2$ consumption measurements (e.g. Lemaitre et al., 2018). It is therefore of interest to investigate the conformity of this transfer function by directly measuring $O_2$ consumption rates and PHP. For that reason, the findings of this study are highly valuable for the community.

Reply: great!

However, the authors report data from only one station (only one data added in both the PHP/$Ba_{xs}$ and JO$_2$-$Ba_{xs}$/JO$_2$-opt correlations) which is weak to support their conclusions. Statistical analyses (p-values, errors on the slopes, etc) are needed.

Reply: we agree that statistical analyses are needed to reinforce conclusions.

Also, a direct comparison of the $Ba_{xs}$/JO$_2$-opt correlation from this study (where the authors show 4 data points; Fig. 3a) with the one proposed by Dehairs et al. (1997) in the Southern Ocean would be very useful and more convincing, to me.

Reply: we added Dehairs data.

Many details are also missing in the methods to really understand how $Ba_{xs}$ concentrations, $O_2$ consumption rates and PHP were measured. Moreover, I would appreciate if there was a discussion about the variations found between ANTARES, PAP and DYFAMED stations, about the differences observed between the Southern Ocean and Mediterranean Sea correlations (Baxs background for example) and about the implications of this study in the water column C budget of the Mediterranean Sea. Finally, all the data ($Ba_{xs}$ concentrations, $O_2$ consumption rates and PHP) should be presented in a Table.

Reply: suggestions are clearly made in the specific comments to improve results presentation and discussion. We'll do our best to answer each of them in the revised ms.

Overall, the manuscript is well written and will be a good fit for publication in Biogeosciences, but considering the lack of details and comparisons, considering the relatively large error bar associated to the JO2-opt, and considering that this study adds only one data point to the JO2 correlation, I would suggest the authors to soften their conclusion on the 'universal validity' of the Dehairs's transfer function.

Reply: we should indeed moderate our conclusion.

**A. Specific comments**

**1-Introduction**
In general, this section should be developed and should mention all the studies on Baxs concentrations as a tracer of POC remineralisation but also all the recent studies investigating barite formation and the role of barite on the Ba cycle

Lines 55-61: Please develop the paragraph about the use of Baxs as a geochemical proxy: more studies have worked on Baxs in the past (e.g. Bishop, 1988; Collier and Edmond, 1984; Ganeshram et al., 2003; Gonzalez-Munos et al., 2003). Moreover, there are some recent studies that should be mentioned/discussed about lab experiments and Ba isotopes giving extremely interesting new insights on the formation of barites and on their role in the Ba cycle in the ocean. Please, see for example the studies of Martinez-Ruiz et al. (2018, 2019), Horner et al. (2015), Cao et al. (2020), Hsieh et al. (2017).

Reply: the present ms. is a "short" paper focusing on the link between Ba and oxygen consumption and on the comparison between SO and MedSea data to test Dehairs equation. That's why we don't go into any more details on isotopes or lab experiments for examples. However, as further requested, few references have been added in the ms.

Lines 66-68: Lemaitre et al. (2018) do not use the Southern Ocean transfer function proposed by Dehairs et al. (1997). These authors proposed a new function specific to the North Atlantic. Consequently, they (sort of) 'revised' the validity of this transfer function at least for the GEOVIDE study area. You could also use this study as an additional reason to check the validity of the Southern Ocean transfer function in the Mediterranean Sea.

Reply: discussion on the relationship reported in Lemaitre et al. is now mentioned and discussed.

Line 67: Instead of Lemaitre et al. (2018), you could also cite Cardinal et al. (2005), Dehairs et al. (2008), Jacquet et al. (2008 a, b, 2011, 2015), Planchon et al. (2013).

Reply: ok added

**2-Sampling and Analyses**
This section must be developed. The reader needs more information and details on how and how well you measure Baxs, $O_2$ consumption rates from optodes and prokaryotic heterotrophic productions. Also, please show all your new data in Figures and Tables. Reply: ok for references and Figures. Tables are not necessary.

Line 115: Please show the full Baxs, pAl and bio Ba depth profile, i.e. from surface to 2000m, in Fig.2a. This will also confirm that the Baxs background stays at around 130pM at depths > 500m.

Reply: sampling was done in the upper 1000 m. We added it section 2.2.

Lines 115-116: If I am correct, the samples used for the data presented in Fig. 2a have not been collected on the same day or exact location. Please prove that there was no evolution of water mass or biology between each sampling. If there was any change, could this influence your Baxs or pAl concentrations?

Reply: Thirteen depths between surface and 1000 m were sampled by combining different casts sampled closeby in time and space (total of 28 samples) and having similar potential temperature – salinity data profiles. No major change in water mass characteristics occurs over the 3-day sampling period (Figure 1c). If there was any change, the risk is that concentrations would reflect another Ba-Al story from a different water mass, e.i. they could reflect an "external" input (lateral, advection, etc…) of particles, or a no local-remineralization-linked signal.

Lines 120-121: Please give the precision and accuracy of your analyses.

Reply: ok added

Lines 124-125: 'sea-salt particulate Ba contribution was found negligible'. What is negligible? Please give numbers.
Reply: done - <0.1%

Line 125: Give more details on the Ba/Al ratio you are using to correct the lithogenic fraction. I suppose it is from the UCC but how does this value compare to the lithogenic inputs at ANTARES? This station is relatively close to the coast and is likely subject to lithogenic inputs, it is therefore important to be sure about the Ba/Al ratio used to correct the lithogenic fraction. Without that, your estimation of Baxs concentrations may not be correct. For comparison, Lemaitre et al. (2018) do not take into account data from two stations where the pBa-litho accounts for 28 and 44% of total Ba. At ANTARES, the Ba biogenic fraction range from 50 to 80%, meaning the lithogenic fraction is not negligible.
Reply: ok we added details on the litho-Ba fraction calculation. We discuss the range of biogenic Ba contribution in session 3.1. The litho impact is negligible at mesopelagic depths (see the grey area in fig. 2a) where it remains <20%. Ba is mostly biogenic at theses depth (>80%).

Line 126: How did you determine the standard uncertainty? From the RSD given by the Element for Ba? From error propagation, taking into account the RSD of Ba and Al?
Reply: yes, we obtain it by error propagation (by taking into account both RSD, uncertainties on Al/Ba ratio, etc…)

Lines 131-136: There is no reference at all in this paragraph – it is thus difficult to understand the technic for someone who is not familiar with this. Please explain, at least briefly, how you measure O2 concentrations with this technic and how you calculate the O2 consumption rates – an equation might help? Can you prove the precision/accuracy of this method? I suppose you need relatively precise measurements to determine an O2 consumption rate. However the errors associated to this measurement and to the final calculation seem high (Fig. 3), why?
Reply: We re-wrote this paragraph. Errors bars seem high because we have 2 optodes per depth. We observe higher variability at upper-mesopelagic depths. Unfortunately we were not able to carry out more than 2 duplicates per depth. We mention it in the ms.

Lines 137-142: Same here, please give more details on the protocol you use for determining the PHP. Why do you use 3H-leucine? How do you then calculate the PHP (equation)?
Reply: the protocol is given in Tamburini et al. (2002). We refer to this paper for detail on protocol and equations.

**3-Results and Discussion**
The authors should give more details to convince the reader about the validity of this Baxs-JO2 function in the MedSea. A direct comparison of the slope of the transfer function you obtain here (Fig. 3a) with the one from the Southern Ocean would be helpful. Some statistics would also help. Moreover, I think this section would get more interesting if there was some comparison with the literature and some explanations on why some of your results slightly differ compared to those of other study areas (essentially, more explanation on the story of the MedSea data – not only about the use of Baxs in this area to trace O2 consumption). The figures could be clearer as well.
Ok

Line 149: 'pAl concentrations are low…' 170nM is not low! On the contrary, it clearly shows a lithogenic input and this makes your Baxs estimations doubtful as the lithogenic correction may not be perfectly constrained. How much is the pBa lithogenic fraction in the depth layer that is interesting for this study (i.e. 100-500m)? Can it be considered as negligible? If yes, why? Please see my previous comment about the Ba/Al ratio and discuss more about the lithogenic correction at

ANTARES station.

Reply: pAl are not low but It is the global lithogenic contribution to the total Ba signal that is low. We corrected the sentence. 170nM was measured in surface. As reported above, we consider that the contribution is negligible at mesopelagic depths. For comparison, pAl measured during the PEACETIME cruise (excluding dust deposits) are in the same range of values at mesopelagic depths as reported here.

Line 156: You mention the pBa biogenic fraction in the interested depth layer is >80% but is it high enough to be assured of a good Baxs estimation? What is the error associated to this correction (this could go to the methods section)?

Reply: yes, it is what is usually assumed. We added details on this correction in the method section.

Lines 157-160: How do you explain the difference between the Baxs background observed in the Southern Ocean and in your study? For example, Lemaitre et al. (2018) also observed a Baxs background at 180pM in the North Atlantic. What is different in the MedSea?

Reply: the difference in Ba background is linked to the saturation level in the MED which is very low compared to other sectors. We discuss it in session 3.1

Lines 165-166: How do you explain the difference of adsorption between ANTARES and DYFAMED stations? Is it related to different bloom timing or intensity?

Reply: it could also be related to a different composition of phytoplanckton material (different species)

Lines 168-169: Please show the full depth profile, i.e. from the surface to 2000m, in Fig. 2a. That would be useful to clearly see the background level.

Reply: sampling was done in the upper 1000 m at ANTARES. We added it section 2.2

Lines 169-170: At DYFAMED station, Baxs concentrations seem to keep decreasing for depths >600m, why is it not stabilised at 130pM?

Reply: The Ba background corresponds to a range of value around which Ba concentrations oscillate at depth.

Lines 180-183: Please, discuss the result of the PAP station if you present it. It is below the trend, why? Moreover, what is the p-value of this trend? Is it a significant correlation with and without the new ANTARES and PAP data? Is it possible to add data from DYFAMED?

Reply: ok. Results at the PAP site reflect a similar situation as observed during KEOPS2 at Plateau site and in a meander of the polar front area (not show in Figure 2b), indicating the temporal evolution and patchiness of the establishment of mesopelagic remineralization processes within a same area. The correlation is reported for KEOPS1, and confronted to KEOPS2, PAP and ANTARES, as given in Jacquet et al. 2015.

Line 184: Are these PHP profiles similar to the one at ANTARES station? Could you plot them all in a figure and add the ANTARES data in a table?

Reply: PHP profiles will add nothing to discussion, because it is ratios of integrated values that are important to be confronted (gradients).

Lines 185-189: 'Indeed, mesopelagic Baxs...' These lines repeat your sentence lines 181-183 '..indicating higher DWA Baxs in situations where a significant part..'. Please re organise this section to avoid repeating things.

Reply: ok done.

Lines 189-190: 'Our MedSea resultS are located..'. You provide only one new result from ANTARES

station, please change the plural to singular form in this sentence. Also, this sentence repeats what you say lines 180-181. Maybe you should delete it.
Reply: ok. We reformulated the sentence.

Lines 190-195: Please develop this section according to the new literature (e.g., Martinez-Ruiz et al., Horner et al., Cao et al., Hsieh et al..) and find a transition with your previous sentence.
Reply: references are added. However, according to personal data (a similar work as Martinez-Ruiz et al. we performed during the BONUS SO cruise) and following Martinez-Ruiz et al., (2018, 2019), it is still unclear, to our understanding, whether barite formation at mesopelagic depths is (directly or indirectly) bacterially induced or bacterially influenced.

Line 200, Fig. 3a and b: It seems that there is a mistake with the units. They do not correspond to those in Jacquet et al. (2015), would it not be mmol/m2/d instead? If I am correct, please change all your JO2 data in umol/L/d and compare the slope you obtain in Fig. 3a with the one from the Dehairs et al. (1997).
Reply: data presented in Jacquet et al. (2015) are integrated values (from 3 to 4 measurements). Each point corresponds to a station. At ANTARES we have only one station (and 4 measurements). This explains the difference of unit. We added a Fig 3c to compare results with Dehairs data;

Lines 201-203: I agree this is a very interesting feature confirming your background Baxs concentration! Could this result give an insight on why there is a different Baxs background in the MedSea compared to other areas?
Reply: as reported above, the difference in Ba background is linked to the saturation level in the MED which is very low compared to other sectors. We discuss it in session 3.1

Line 209: Why do you use a factor of 17450 here while it is 17200 in Jacquet et al. (2008) or Lemaitre et al. (2018)?
Reply: according to data presented in Dehairs et al., 1997, the ratio is 17450. We added it in Fig3c. No idea why 17200 is used elsewhere (we removed points in the correlation?).

Lines 214-219: There is a large error bar associated to your ANTARES data point for JO2-opt (Fig. 3b), why? I agree that considering this large error bar, your data fits the trend observed during KEOPS. However, this large error bar and the poor distributions of the data points (either low JO2 for KEOPS or high JO2 for ANTARES) make this correlation too weak to state that there is no difference between both regions. What is the p-value of this correlation with and without ANTARES? Is it possible to add data from PAP or DYFAMED stations? I would be more convinced by a comparison of your Baxs-JO2 trend with the one of the Southern Ocean. For now, the slope in Fig. 3a is very different from the one of the Southern Ocean (100 versus 17450). After fixing the unit problem, please discuss about this comparison.
Reply: data are not available for PAP or DYFAMED (no JO2 measured). As reported above the error bar is obtained form error propagation following Dehairs equation. Comparison with Dehairs's data is given in Figure 3c.

Line 226: Please indicate what is Z in this study.
Reply: ok 175-450 m.

Lines 228-229: Please give the range of the fluxes from the literature and discuss them according to the one you estimate at ANTARES.
Reply: done

Lines 239-241: Expand a bit the discussion here. How does your study contribute to the MedSea carbon budget? Does it help balancing the water column budget?

Reply: done

**B. Line notes**

Abstract:
Lines 25-27: These are not new observations/conclusions. Please make it clear here that you are confirming what has been observed earlier in another area (Jacquet et al., 2015).
Reply: done

Line 25: 'higher Baxs (409 pM; 100- 500 m) [occurs] in situations where integrated PHP (PHP100/500= 0.90) is located deeper'
Reply: ok added

Line 26: 'higher Baxs [occurs] with increasing JO2-Opt'
Reply: ok added
Introduction:

Line 63: 'highly resolved, precise..' seems a bit exaggerated as a sampling resolution of 50m depth is good but not high for me and I suppose the technics may be more precise today compared to 1997.
Reply: ok deleted

Line 70: I would delete this sentence as it repeats the sentence line 68 ('Yet its validity has never been tested..') and it separates two linked sentences.
Reply: ok deleted

Line 70: 'These advancements..' refer to the results of Jacquet et al. (2015) I suppose? Please make it clearer.
Reply: ok

Line 83: Which fluxes are you referring here? Primary production, export, remineralistion?
Line 83: Please give a range of the fluxes determined by Santinelli et al (2010) and Ramondec et al. (2016).  Reply: remineralization and fluxes are given in discussion

Methods:
Line 111: I would name this whole section 'Methods' and would name the sub-section 2.2 'Sampling and Analyses'
Reply: OK done

Line 107: 'and [(3)] Levantine Intermediate Water…'
Reply: ok added

Line 117: 'total digestion of filters using a [concentrated] tri-acid mixture..'
Reply: ok added

Line 130: 'The background (or residual value) is considered as "preformed" Baxs at zero oxygen consumption left over after transfer and partial dissolution of Baxs produced during degradation of previous phytoplankton growth events. [The background is set at 130pM in this study].'
Reply: ok added

Results and Discussion:
Line 145: Maybe modify to '[Particulate Baxs] vertical disctribution' to avoid any confusion for the reader.

Reply: ok done

Line 160: 'For comparison, the [Baxs] background value…'
Reply: ok added

Line 173: 'the particulate excess Ba (>BKG)' is confusing for me. You never expressed Baxs like this
before. Please keep the same wording all along the manuscript, maybe modify to 'The maxima Baxs
concentrations are centred..'
Reply: ok modified

Line 174: 'in this depth layer' instead of 'at these depths'
Reply: ok modified

Lines 174-175: Explain what is the depth-weighted average, as you did for example in Jacquet et al.
(2015): 'i.e. the Baxs inventory divided by the depth layer considered Z'.
Reply: ok added

Line 175: 'over the 100-500m depth layer' instead of 'this entire depth layer'. It will avoid any
confusion with Fig.2b and all the different depth integrations.
Reply: ok

Line 176: Figure 2b shows [the] column-integrated PHP at 100m over the [one] at 500m
(PHP100/500). Our PHP100/500 ratio at ANTARES station is of 0.90 and is compared to results
obtained during KEOPS1…'
Reply: ok modified

Line 180: 'ResultS at the ANTARES..' Are there more than one result? On Fig.2b, there is only one
data from ANTARES station.
Reply: yes there is only one point because it is an integrated data for 1 unique station

Line 181: '…follow the trend previously reported in the Southern Ocean [(blue dashed line in Fig.2b;
Jacquet et al., 2015)]..'
Reply: ok

Lines 181-182: Please make it clear that the ANTARES data confirms the conclusions found in Jacquet
et al. (2015) and that it is not a new conclusion.
Reply: ok done

Line 204: '[In Figure 3b,] we applied..'
Reply: ok added

Line 217: 'Overall, our results indicate [a] similar Baxs-JO2 relationship..'
Reply: ok added

Lines 257-258: You also show the DYFAMED station in this figure. Please mention it is for comparison
and cite Sternberg et al. (2008).
Reply: ok added

Line 258: '[c] potential temperature-salinity-depth plots…'
Reply: ok

Line 269: Could you integrate the DWA Baxs between 100-500m as well (to match with the PHP

integration)?
Reply: it is a mistake in the text; integration is done at 100m and not 150m. We corrected it.

Lines 269-271: 'Regression of the same ratio is reported for KEOPS1 ([light blue symbols;] out plateau stations) and KEOPS2 ([dark blue symbols;] Southern Ocean; Jacquet et al., 2015) and #DY032 ([red square;] PAP station, NE-Atlantic; pers. data) cruises.'
Reply: ok

Lines 269-271: Please clarify what the blue dashed line represents. Is it from Jacquet et al. (2015) or does it take into account all data points including the new ones from ANTARES and PAP stations?
Reply: It is KEOPS2 only. We specified it in Fig2.

Line 275: mmol/m2/d instead?
Reply: no, it is the correct unit umol /L/d

Lines 275-276: '..optode measurements (this study; [green square]), dark community respiration DCR (winkler titration; [red triangles]; JO2-DCR; Jaquet et al., 2015; KEOPS1)'
Reply: ok added

Lines 277-278: It is not clear if you speak about the y-axis or the black line. I propose to re write as '…and [Baxs contents (Southern Ocean transfer function from Dehairs et al. (1997); JO2-Ba]. The black line corresponds to the correlation found in Jacquet et al. (2015)'. If this is correct, please also mention that this correlation excludes some data points from A3 and E stations.
Reply: ok modified

Figure 2a, in the legend: Ba[xs] ANTARES; Ba[xs] DYFAMED; [p]Al ANTARES. Reply: ok And please show the full depth profile (until 2000m). Reply: profiles are limited to 1000m depth.

Figure 2b: Please indicate from where the blue line comes from. And indicate the p-value.
Reply: done

Figure 3a: Please check the units and indicate JO2 in umol/L/d. And show the trend from Dehairs et al. (1997) in the Southern Ocean. Give the p-value.
Reply: units are correct. Done

Figure 3b: Please indicate JO2 in umol/L/d. Also, indicate from where the black line comes from. And indicate the p-value.
Reply: ok

**C. References:**
Reply: thank you for the references

---

## Author Comment (AC2) · 1 Dec 2020

On the barium-oxygen consumption relationship in the Mediterranean Sea: implications for mesopelagic marine snow remineralization

Authors: Jacquet et al.

Response to Referee #2

The authors present new data concerning the relation between biogenic barium (Baxs), the O2 consumption and prokaryotic heterotrophic production (PHP) in the Mediterranean Sea. The purpose of this paper is to improve our understanding of the relation between barium and oxygen and to test the validity of the Dehairs transfer function in the Mediterranean Sea. This relation has never been tested in the Mediterranean Sea. They also investigated further the relation between PHP and Baxs distribution. I think the paper has nicely approached these issues with their new dataset. Although I think the dataset and the statistics of the study are weak and the paper is missing some important information.
Reply: As also reported by Referee #1, we agree that statistical analyses are needed to reinforce the ms.

Nevertheless, such information is still valuable for the community and may help to improve our understanding of barium cycle in the ocean. I would recommend the manuscript for publication in Biogeosciences.
Reply: great!

However, I list issues below, which I think the authors should consider in their revision: My main concern for this paper is that the authors conclude that there is strong relationship between Baxs and JO2 rates and that the transfer function can be apply with no restriction in the MedSea. The authors should be more moderate about these statements considering that there are not that many data and the lack of statistical analysis for these relationships.
Reply: we should indeed add statistical analysis and moderate/reformulate our conclusion.

Indeed, linear regressions in figures 2b), 3a) and b) should take into account the errors bars. The errors on the slope and intercept should be shown, as well as the p value to show if the relations are significant.
Reply: error bars and p value are added in Fig. 2 and 3

On figure 2a), only data from KEOPS 2 are considered for the regression. The regression should take all the data (KEOPS 1; KEOPS 2 and PAP). Error bars of these data should be taking into account. Then, a 95% confidence interval could also be added to show that the ANTARES data point is in that interval.
Reply: the aim is to compare KEOPS1 regression and our new MEDSEA data. KEOPS2 data are compared to KEOPS1 in Jacquet et al., 2015.

Concerning the JO2 from optode vs JO2 from Baxs (Figure 3a and the associated paragraph (lines 198–203)), the fact that the intercept matches the background is an interesting feature. However, this feature is biased by the fact that the regression is taking into account the value at 1000m (130pM). Indeed, this value from 1000m is used as the background and then use in the regression to prove that the background is close to 130pM. It is a circular reasoning. Indeed, this value (1000m≃30pM) forces the regression and so should not be used for that regression. The regression should take only value at 175m, 250m and 450m. The error bars for these values should also be taking into account in that regression. Errors on the slope and intercept should be provided especially if you are discussing the fact that the intercept match the background value.
Reply: even if we remove the value at 1000 m, bkg reaches a very close value, not significantly

different, i.e. 141 pM. 130 pM is an artibrary value, taken looking at profiles shape (i.e. value reached below 500 m at DYFAMED and ANTARES). It is reasonable to keep it in the regression.

In this figure, it will also be interesting to see the data from the Southern Ocean (Dehairs et al., 1997) and the North Atlantic (Lemaitre et al., 2018) as a comparison.
Reply: comparison with Dehairs data is now done in Fig3c.

For the JO2 Ba vs JO2 measured relationship (figure 3b), the authors say that MedSea data are 3 times higher than KEOPS data. And there is only one point for the MedSea with important error bars. Considering all of that it seems hard to say that the MedSea show the same relationship than the Southern Ocean and even more saying that this support the universal validity of the Dehair's transfer function. Maybe a 95% interval would be useful in this figure too. This interval would show that the ANTARES value is good agreement with the relationship from KEOPS data. More data would be needed to state the universal validity of the Dehair's transfer function.
Reply: we re-worked on correlations, and provided statistical analyses. We added missing errors bars and comparisons between med Sea and SO data. We also added discussion on Background values.

Concerning the analyses part, different information is missing. First, only few information is provided on how pAl data have been generated. The authors should provide more information on the sampling, the analysis of these data and their accuracy.
Reply: the sampling and analysis parts have been completed with more details.

Moreover, the authors should elaborate why and how pAl used to correct Ba from the lithogenic fraction would help the reader. The authors do not provide any references for the measurement of the O2 consumption rates. More explanations and references are needed to help the reader understand how these data have been generated. Please also explain how from oxygen concentrations you obtain the consumption rates (linear model calculations), maybe with equation. Provide the accuracy of these data. In the same way, more information and references on PHP measurements and why PHP are interesting to compare to Ba and O2 (in the introduction) will make the rest of manuscripts easier to understand for the reader. Also the accuracy these data should be provided. In the manuscript and figures, different units are used the O2 consumption data, please verify and unify.
Reply: we added the necessary references for Al corrections, for O2 measurements and calculations, as well as for PHP. Units have been verified.

Finally, the data are never shown in tables, data should be presented in tables in the manuscript or at least in supplementary materials.
Reply: as also reported to referee #1, a supplementary table is not necessary and Figures have been completed.

---

## Author Response (AR2)

On the barium-oxygen consumption relationship in the Mediterranean Sea: implications for mesopelagic marine snow remineralization
Authors: Jacquet et al.

Response to Referee #1 – Second Round

Section 2.2. Sampling and Analyses.
• Line 134: Precise in which section you will discuss about the biogenic Ba fraction in more details. 'Section 3.1' instead of 'see below'.
Reply : Ok, modified.

• Lines 151-156: The method for measuring PHP is still not explained. It would make it a lot easier for the reader if it was briefly explained, even if it is the same method than in Tamburini et al. (2002).
Reply : according to co-authors who wrote this section, it is not necessary to further develop on the method. It is sufficiently summarized. Details can be found in Tamburini et al. (2002).

Section 3.1 Particulate Baxs vertical distribution
• Lines 163-164 and lines 169-170: Could you please mention that the main focus of this manuscript is on the mesopelagic layer and that having the biogenic Ba greater than 80% in this layer is essential for neglecting the lithogenic fraction here? As it is now, reading that the biogenic Ba range between 50 and 90% - or instead that the lithogenic fraction is up to 50% - still gives doubt on the Baxs estimation. Please, include in the text what you said in your replies.
Reply : ok, added.

Section 3.2 PHP
• Lines 195-198 (Figure 2b): I wonder how does the KEOPS2 correlation compare with a global correlation (ie, taking into account all data points, including KEOPS1, KEOPS2, ANTARES and PAP data)? Maybe you could use this global correlation instead of constraining to only KEOPS2? This would generalise your conclusions.
Reply : KEOPS1 is used as a reference. The aim is not to define a single correlation combining the different cruises but to compare different seasonal situations in a same sector and understand potential evolution. That is why it is non-sense to generalise a single correlation.

Section 3.4 Estimated particles remineralisation rates and implication
• Lines 253-259: It is not clear to what Ba fluxes correspond to. Are they downward export fluxes, Ba release fluxes? Why are the Ba fluxes in ANTARES so much lower than in DYFAMED or in the Alboran Sea?
Lines 260: Are the fluxes from Speicher et al. (2006) from the Mediterranean Sea as well? Please, precise the location and also the POC fluxes (give numbers).
Reply : mesopelagic remineralization (100-1000m) = difference between surface production (upper 100 m) and deep (below 1000) export. Differences could be due to the season stage and to the particles collection system used (bottles vs sediment traps) or type of calculation (Th-derived data, profile integration…). We already noticed in the ms. the associated method to fluxes reported.

• Lines 263-264: You say 'remineralisation rates in the Mediterranean Sea is far from being achieved' but what does it mean? Is it about the spatial resolution, the seasonal changes, the global understanding? What else has to be done?
Reply : Ok, explained. Inter-basins variability and season advancement are the main questions.

---

## Author Response (AR3)

On the barium-oxygen consumption relationship in the Mediterranean Sea: implications for mesopelagic marine snow remineralization
Authors: Jacquet et al.

• **Associate Editor Decision: Publish subject to minor revisions (review by editor)** (14 Feb 2021) by Carolin Löscher
Comments to the Author:
Dear authors
thanks for your re-submission and response. The reviewers repeatedly asked for describing the method for measuring PHP. I understand it is described elsewhere, still, please provide a short description, it is not beneficial for the reading flow to jump to another paper.
All the best
Carolin Löscher

Reply : Method for measuring PHP is now better explained.

"Prokaryotic heterotrophic production (PHP) estimation was measured over time course experiments at in situ temperature (13°C) following the protocol described in Tamburini et al. [2002]. 3H-leucine labelled tracer [Kirchman, 1993] was used. For water sample collected with Niskin bottle we have performed measurement in three replicate 20 mL and 40 mL seawater volume for the depth ranged 0 to 800 m-depht with 20nM at final concentration of Leucine. Concerning depth above 800 m-depth, PHP was measured in three replicate of 40 mL of seawater with 10nM at final concentration of Leucine. Samples were incubated 2, 6 and 10 hours respectively for sample ranged between 0-200 m, 200-600 m and up 800 m-depth. To calculate prokaryotic heterotrophic production, we used the empirical conversion factor of 1.55 ng C per pmol of incorporated leucine according to Simon and Azam [1989], assuming that isotope dilution was negligible under these saturating concentrations."